# Exosomes in Glioma: Unraveling Their Roles in Progression, Diagnosis, and Therapy

**DOI:** 10.3390/cancers16040823

**Published:** 2024-02-18

**Authors:** Song Yang, Yumeng Sun, Wei Liu, Yi Zhang, Guozhu Sun, Bai Xiang, Jiankai Yang

**Affiliations:** 1Department of Neurosurgery, The Second Hospital of Hebei Medical University, Shijiazhuang 050000, China; 2Department of Immunology, College of Basic Medicine, Hebei Medical University, Shijiazhuang 050017, China; 3Department of Cancer Genetics and Epigenetics, Beckman Research Institute, City of Hope Cancer Center, Duarte, CA 91010, USA; 4College of Pharmacy, Hebei Medical University, Shijiazhuang 050000, China

**Keywords:** exosomes, gliomas, diagnosis, treatment, drug delivery system

## Abstract

**Simple Summary:**

Glioma is the most prevalent form of primary malignant brain tumor and is often associated with a grim prognosis. Exosomes, minute extracellular vesicles, are secreted by various cells and hold a significant role in tumor growth and invasion. Examining the contents they carry allows for a deeper understanding of tumor characteristics and progression. Moreover, exosomes possess the capability to traverse the blood–brain barrier, functioning as effective delivery vehicles. By encapsulating chemotherapy drugs and other molecules, we can achieve an enhanced therapeutic outcome. This article primarily delves into the impact of exosomes on glioma development, as well as the latest strides in diagnosis and treatment. This underscores the advantages and potential avenues for future research in the realm of exosomes for glioma diagnosis and treatment.

**Abstract:**

Gliomas, the most prevalent primary malignant brain tumors, present a challenging prognosis even after undergoing surgery, radiation, and chemotherapy. Exosomes, nano-sized extracellular vesicles secreted by various cells, play a pivotal role in glioma progression and contribute to resistance against chemotherapy and radiotherapy by facilitating the transportation of biological molecules and promoting intercellular communication within the tumor microenvironment. Moreover, exosomes exhibit the remarkable ability to traverse the blood–brain barrier, positioning them as potent carriers for therapeutic delivery. These attributes hold promise for enhancing glioma diagnosis, prognosis, and treatment. Recent years have witnessed significant advancements in exosome research within the realm of tumors. In this article, we primarily focus on elucidating the role of exosomes in glioma development, highlighting the latest breakthroughs in therapeutic and diagnostic approaches, and outlining prospective directions for future research.

## 1. Introduction

The World Health Organization (WHO) identifies over 100 distinct types of central nervous system (CNS) tumors, categorized by their origin, location, and histological traits [1]. Notably, their unique molecular components are now utilized for classification, signaling a forthcoming trend in precise molecular medicine. Gliomas represent the most prevalent category of primary intracranial tumors, comprising 31% of all brain and CNS tumors and 81% of malignant brain and CNS tumors [2]. Key features of gliomas include multiple occurrences, marked by a high growth rate, extensive invasion, and genetic variations [3].

Glioblastoma (GBM), comprising 70% of all gliomas, exhibits a median overall survival of merely 15 months [4]. This malignancy is distinguished by nuclear atypia, increased mitotic activity, microvascular proliferation, and tissue necrosis [5]. Presently, the Stupp protocol following extensive surgical resection, encompassing radiotherapy and chemotherapy with temozolomide, represents the global standard of care. However, its efficacy remains limited due to challenges associated with surgical procedures, inadequate medication penetration across the BBB, histological heterogeneity, and the aggressive nature of GBM [6]. Despite advancements in combination therapy leading to improved progression-free survival, overall survival rates remain largely unchanged [4]. As a result, current glioma treatment is primarily focused on extending the patient’s lifespan rather than achieving a definitive cure.

The BBB constitutes a specialized biological barrier, establishing a tightly regulated neurovascular unit (NVU) that governs the CNS’s homeostasis [7]. This NVU comprises a basal barrier supported by pericytes and astrocytes, with specialized endothelial cells (ECs) connected by tight junctions (TJs). In addition to physical barriers, a “metabolic barrier” operates to swiftly transform undesirable metabolites, peptides, hormones, and other substances that enter the brain parenchyma [8]. Furthermore, a range of multi-substrate efflux pumps actively transport undesired compounds back to the luminal side, including the majority of medications targeting neurological conditions [9]. Complexity arises from substantial BBB heterogeneity across different brain regions, variations in delivery methods, and alterations induced by disease states [10,11]. Thus, the BBB, despite its name, functions as a multi-faceted interface. The BBB’s constraints restrict the clinical use of only a handful of chemotherapeutic agents, including Temozolomide, for glioma treatment [12]. However, their impact on overall survival is markedly limited. Various methods have been devised to increase drug concentrations in the brain, including the use of appropriate delivery vehicles, biodegradable wafers and techniques involving ultrasounds, convection-enhanced delivery (CED), and vasoactive compound injections [13,14,15].

Leveraging nanoparticles (NPs) as delivery vectors represents a promising strategy for achieving effective and targeted drug delivery to tumor sites [16]. Their advantageous physicochemical characteristics, self-assembly capability, bio-compatibility, and customizable surface decoration for tumor targeting contribute to their significant potential in addressing GBM [17,18]. Moreover, NPs enable controlled drug release, thereby minimizing the necessary dosage frequency [19]. While nanotechnology-enabled drug delivery has shown encouraging results since its inception, there remains a demand for non-toxic yet highly effective delivery methods, given the synthetic nature of these nanocarrier systems, which poses potential toxicity challenges within the body [20].

In this context, extracellular vesicles (EVs), particularly exosomes, emerge as promising candidates to meet this demand. Exosomes, nano-sized extracellular vesicles released by various cells, demonstrate exceptional efficacy in traversing the BBB, establishing them as potent carriers for therapeutic delivery [21]. Additionally, exosomes play a pivotal role in glioma progression by facilitating the transportation of biological molecules and promoting intercellular communication within the tumor microenvironment [22,23]. These attributes offer promise for improving glioma diagnosis, prognosis, and treatment. This review centers on elucidating exosome characteristics, their involvement in glioma development, showcasing recent breakthroughs in therapeutic and diagnostic approaches, and outlining future research directions.

## 2. Exosomes’ Biology and Potentiality for Crossing the BBB

### 2.1. Production and Structure of Exosomes

Cells, both prokaryotes and eukaryotes, release EVs as part of their regular physiological processes and in response to acquired abnormalities [24]. EVs are membrane-enclosed vesicles, carrying various substances that influence a wide range of cellular functions through the transmission of active genetic, protein, or lipid-based instructions [25,26]. Exosomes, microvesicles, and apoptotic bodies are common types of EVs, with autophagic EVs, stressed EVs, and matrix vesicles being more recently identified [27]. Exosomes, ranging from 40 to 150 nm in diameter, originate from endosomes through biogenesis, transport, and release processes [24,28]. The term “exosome” is derived from “endosomes” and “secreted via exocytosis”, although it is recommended to refer to them as small EVs (sEV) due to the lack of specific characteristics of exocytosis in many cases [29].

Exosomes are characterized by a bilayer membrane and carry a diverse cargo, including proteins, lipids, enzymes, transcriptional factors, DNA fragments, mRNAs, micro-RNAs, and long non-coding RNAs (lncRNA) [24,30]. They are released by various cell types, including cancer cells, lymphocytes, dendritic cells (DCs), adipocytes, and fibroblasts. Exosomes have been found in biofluids such as blood, plasma, urine, cerebrospinal fluid (CSF), milk, amniotic fluid, malignant ascites, saliva, and synovial fluid. They facilitate the transfer of materials between donor and recipient cells, playing a vital role in both normal and pathological cellular communication [23,31].

### 2.2. Isolation Methods of Exosomes

Exosome isolation techniques currently in use encompass various methods, including ultracentrifugation, density-based separation, precipitation, ultrafiltration, and immunoaffinity [32]. These techniques primarily rely on the physical characteristics of exosomes, such as their particle size and density. Ultracentrifugation is widely regarded as the gold standard for exosome separation due to its high yield, consistent purity, and its ability to meet the demands of future biochemical and cellular research [33]. Ultrafiltration, relying on molecular size, stands as a straightforward approach for exosome separation [34]. He et al. introduced a highly efficient optimized ultrafiltration method, adept at eliminating impurities larger than 200 μm and reducing the concentrate to 1/50 of the original volume [35]. In recent years, the polymer precipitation method, traditionally employed for isolating viruses or other biological macromolecules, has gained popularity in exosomal isolation [33]. However, exosomes obtained through this method are prone to contamination with lipoproteins or virus particles, potentially impacting subsequent analyses like proteomics and mass spectrometry [36]. Ongoing exploration and discussion are focused on emerging methods for isolating exosomes [37].

However, distinguishing exosomes from other extracellular vesicles (EVs) can be challenging [38]. In an effort to address this, Ibsen et al. have developed an electrokinetic microarray chip technology that operates on alternating current and requires only 30 to 50 µL of plasma, with a processing time of just 15 min [39]. As an example, Cumba Garcia et al. employed practical density gradient-based ultracentrifugation to isolate exosomes from the plasma of patients with glioblastoma (GBM) for biomarker analysis [40]. Additionally, it is feasible to ascertain both the presence and purity of exosome preparations via verification of exosome-specific markers (CD9, CD63, ALIX, or TSG-101) by, for example, western blotting [40].

### 2.3. Potentiality for Crossing the BBB Based on Exosomes

Nanocarriers, due to their small size and adaptability, have shown significant promise in drug delivery. Exosomes, as natural nanoparticles, have garnered attention for CNS disease treatment, owing to their potential for natural BBB crossing and versatile surface engineering. Banks and colleagues investigated the capacity of 10 exosome populations obtained from mouse, human, cancerous, and non-cancerous cell lines to traverse the blood–brain barrier [41]. The study revealed that all tested exosomes successfully crossed the blood–brain barrier, exhibiting diverse rates and engaging in various vesicular-mediated mechanisms, including specific transporters, adsorptive transcytosis, and a brain-to-blood efflux system. [42] Recent research indicates that exosomes derived from different parent cells exhibit varying tropisms for organs and tissues [43]. For example, neural-stem-cell-derived EVs demonstrated enhanced CNS distribution compared to mesenchymal-stem-cell-derived EVs in a mouse stroke model [44]. Furthermore, under specific pathological conditions, such as brain inflammation or tumors, there is a notable enhancement in exosome transport across the blood–brain barrier [45,46]. In mice with brain inflammation, exosomes derived from macrophages demonstrated more than a three-fold increase in delivery to the brain compared to those in normal mice [47].

Studies have postulated two possibilities for the uptake of exosomes by the brain: either being sequestered inside the brain endothelial cells or undergoing complete passage through the endothelial cell barrier [41,45]. Julien Saint-Pol et al. propose several hypothetical pathways to elucidate the communication and interaction between exosomes and the target brain cells [48]. Through the utilization of uptake mechanisms, strategies have been employed to enhance the potential of exosomes to traverse the blood–brain barrier as follows.

The process of receptor-mediated transcytosis (RMT) involves the movement of invaginated endosomal compartments to the opposite side of the membrane and is initiated by the binding of a specific ligand to its corresponding receptor. RMT has been extensively studied and widely applied for transportation through the endothelial cells of the blood–brain barrier [49]. Various therapeutics, such as chemicals, antibodies, polymers, and nanoparticles, can incorporate these strategies [50]. Similarly, exosomes can leverage RMT by generating specific ligands that bind to receptors involved in transcytosis. For example, Kim et al. used a T7 peptide (T7-exo) to deliver exosomes [51]. The T7 peptide, with the HAI-YPRH sequence, binds to the transferrin receptor (TfR) without interfering with transferrin’s ability to bind to TfR [52]. When administered intravenously, T7-exo exhibited enhanced targeting of intracranial glioma in rat models compared to unmodified exosomes. Li et al. employed the angiopep-2 peptide to modify exosomes, showcasing its capability to induce endocytosis and enhance the ability to traverse the blood–brain barrier [53] (Figure 1). Moreover, the pathway of modifying exosomes to bind to receptors such as the insulin receptor (INSR) for enhancing RMT awaits exploration and validation [50].

Additionally, exosome modification can involve peptides that bind to specific membrane proteins. For instance, the c(RGDyK) peptide, which binds to the highly expressed integrin αvβ3 in brain capillary endothelial cells (BCECs), has been used to enhance exosome targeting. In a mouse stroke model, exosomes labeled with this peptide exhibited an 11-fold increase in transport to the ischemic area of the brain compared to scrambled peptide-labeled exosomes [54]. Additional findings also indicate that cRGD-modified exosomes loaded with PTX significantly enhance the therapeutic effects of PTX in GBM through improved targeting [55].

Neurotropic viruses possess the capability to traverse the BBB and infiltrate the brain parenchyma, sparking research into the utilization of viruses or viral components as carriers for delivering therapeutics to the brain [56]. For instance, peptides derived from the glycoprotein of the rabies virus (RVG) demonstrate efficient penetration through the BBB and targeted delivery to neurons [57]. Numerous preclinical studies have investigated the application of RVG-derived peptides to facilitate brain-specific targeting of exosomes [58]. Section 5.2 of this review delves into the application of exosomes’ BBB-penetrating potential as delivery vectors.

## 3. Role of Exosomes in Glioma Progression

Exosomes play a vital role in facilitating cell-to-cell communication by transporting bioactive molecules from donor to recipient cells [22]. Studies have demonstrated that cancer cells, including gliomas, release exosomes abundantly to communicate with neighboring cells both locally and distantly [23]. Glioma-derived exosomes (GDEs) exert influence on various aspects of tumor development and progression, including the establishment of the pre-metastatic niche, immune evasion, angiogenesis, anti-apoptotic signaling, and resistance to treatment. Conversely, exosomes from healthy cells, such as T cells, B cells, and dendritic cells, have been shown to significantly inhibit tumor growth [59]. This multifaceted impact is attributed to various proteins, mRNAs, and noncoding RNAs carried by exosomes (Figure 2).

### 3.1. Effect on the Microenvironment around Gliomas

The tumor microenvironment (TME) in gliomas encompasses a diverse array of both cancer and non-cancer cells, including endothelial cells, immune cells, glioma stem-like cells, and non-cellular components like the extracellular matrix [60]. The TME has emerged as a powerful driver of glioma progression and a critical regulator of tumor growth [61]. Exosomes have been identified as crucial mediators of communication between the tumor and the TME. For instance, a study revealed that the miR-340-5p-macrophage feedback loop altered both GBM development and the TME [62]. Another study demonstrated that tumor-suppressive miR-3591-3p could be released via exosomes to target tumor-associated macrophages in glioma cells, thus creating a suppressive microenvironment that facilitated glioma invasion and migration [63].

### 3.2. Angiogenesis Mediated by Exosomes

Exosomes released by gliomas play a pivotal role in driving angiogenesis, a critical process in glioma progression. For example, a study investigating the impact of glioma cells on angiogenesis found that glioma cells could enhance this process by delivering Linc-CCAT2 to endothelial cells via exosomes [64]. Additionally, research by Li Yan et al. highlighted the significance of exosome-derived circGLIS3 in high-grade glioma, where it contributes to glioma invasion and angiogenesis by regulating Ezrin T567 phosphorylation [65]. Notably, ECRG4-EV generated from brain endothelial cells has been shown to inhibit glioma growth by regulating angiogenesis and inflammation [66].

### 3.3. Role of Exosomes in the Proliferation and Invasiveness of Gliomas

Glioma-derived exosomes (GDEs) contain both coding and noncoding RNA elements, rendering them pivotal in regulating tumor growth, survival, and invasion, crucial aspects of glioma cell persistence and recurrence [67]. For instance, the presence of exosomal miR-148a, known to target CAMD1 and boost signal transducer and activator of transcription 3 (STAT3) activity, has been linked to the advancement of glioblastoma multiforme (GBM) [68]. Studies have confirmed that both overexpression of miR-486-5p and silencing of circZNF652 hindered cell proliferation, migration, invasion, and epithelial-mesenchymal transition (EMT) in GBM cells. Notably, GBM-cell-derived exosomes were characterized by highly expressed CircZNF652 [69]. Furthermore, research has demonstrated that miR-3184-3p, circARID1A, and miR-9, abundant in GDEs, fostered the proliferation, invasion, and migration of glioma cells [70,71,72].

### 3.4. Role of Exosomes in the Resistance to Glioma Treatment

Exosomes play a pivotal role in the resistance exhibited by gliomas against treatment, as they can transport drugs out of tumor cells [73]. They may stimulate the development of fibroblasts that lead to fibroblastic responses, acting as a barrier against anti-tumor medications. Additionally, through biomolecules such as miRNA, exosomes can transform drug-sensitive tumor cells into drug-resistant ones [74,75]. Furthermore, hypoxia induces specific signaling pathways, bolstering glioma resistance to treatment through exosome absorption [76]. According to Garnier et al. [77], glioma stem cells (GSCs) release exosome-transcribed MGMT mRNA, reflecting TMZ resistance. Another study established that MGMT genomic rearrangements contribute to TMZ resistance by repairing the O6-Methylguanine lesion caused by TMZ [78]. Moreover, research has highlighted the potential of the exosome-mediated circWDR62 and macrophage migration inhibitory factor (MIF) in increasing TMZ resistance in glioma, suggesting their value as prognostic biomarkers [79,80]. This also emphasizes the involvement of tumor macrophages in recurrent GBM [81].

### 3.5. Exosomes in Mediating Immune Responses

Exosomes derived from tumors interact with various immune cells, including effector T cells, naturally occurring T_reg_ cells, and natural killer (NK) cells, contributing to immune suppression and tumor growth [82]. Antigen-presenting molecules, TGF-b, tumor antigens, and immune intracellular adhesion molecules can all be identified in the serum exosomes separated from GBM patients [83]. Brain tumor-initiating cells produce the extracellular matrix protein TNC associated with exosomes to limit T cell activity, as demonstrated in research by Reza and colleagues [84]. TNC interacts with the α5β1 and αvβ6 integrins on T cells, leading to reduced mTOR signaling and halting T cell growth. Another study revealed that miR-1246 mediates H-GDE-induced M2 macrophage polarization by targeting TERF2IP, activating the STAT3 signaling pathway while hindering the NF-B signaling pathway, potentially creating an immunosuppressive microenvironment [85]. A recent study found that circNEIL3, packaged into exosomes by hnRNPA2B1, is conveyed to infiltrated tumor-associated macrophages, allowing them to acquire immunosuppressive properties by stabilizing IGF2BP3, thereby promoting glioma progression [86].

In summary, the bioactive molecules transported by exosomes play a significant role in glioma progression. It is important to note that the various stages of tumor development are interconnected and mutually influenced, rather than independent. Further exploration of the types and mechanisms of action of bioactive molecules influencing glioma progression is essential for their potential use as diagnostic and therapeutic tools.

## 4. Exosomes as a Promising Strategy for Diagnosis and Prognosis of GBM

The current standard of diagnosis for GBM patients involves MRI, surgery, or brain biopsies [87]. However, these strategies have limitations. MRI may miss subtle lesions due to its limited sensitivity and precision. Additionally, distinguishing between tumor recurrence and postsurgical necrotic areas can be challenging, and pinpointing the exact tumor location without histological examination is difficult [88]. Obtaining histology samples through direct surgery or biopsies is also challenging due to surgical risks and the heterogeneous nature of the tumor, making it less reliable as a one-time intervention. Unlike other cancers, diagnosing brain tumors using circulating biomarkers like circulating tumor cells and cell-free nucleic acids is hindered by the blood–brain barrier [89].

As discussed earlier, exosomes play a crucial role in tumor development, making them a logical target for tumor detection and monitoring based on their contents. Exosomes have been extensively studied to assess their potential as diagnostic and prognostic biomarkers.

### 4.1. MiRNA

MiRNAs regulate post-transcriptional gene expression, and dysfunctions in these molecules can affect cellular physiology and development, leading to disease [90]. Exosomes from GBM patients have been shown to contain a significant number of miRNAs [91]. MiRNAs, being highly stable within exosomes and exhibiting differential expression depending on the tumor type, make promising biomarkers for the diagnosis and prognosis of malignancies [92].

One of the early miRNAs proposed for diagnostic use in GBM patients was exosome-derived miR-21. MiR-21, frequently overexpressed in malignant gliomas, modulates glioma cell invasiveness, migration, and tumor grades [93,94]. The proportion of miR-21 in exosomes from the CSF of GBM patients was found to be ten times higher than in exosomes from healthy participants. Similarly, miR-21 levels were 40 times higher in the serum exosomes of GBM patients compared to healthy individuals [95]. The use of exosomal miR-21 for glioma diagnosis showed good effectiveness according to receiver-operating characteristic curve studies [96]. Serum exosomes from high-grade glioma patients were found to encapsulate significantly higher levels of miR-21, miR-222, and miR-124-3p when compared to both healthy controls and low-grade glioma patients.

Moreover, subsequent reports by Santangelo et al. in 2018 indicated a rapid decrease in these miRNA levels following neurosurgical treatment [97]. Utilizing minimally invasive techniques, it becomes feasible to detect the glioma-derived exosome (GDE) contents of miR-21, miR-222, and miR-124-3p, facilitating the identification of brain tumors, prediction of glioma pathological grading, and the discovery of preoperative metastases [97,98]. In a separate study, exosomal miR-181 emerged as a promising biomarker for early-stage human gliomas [99]. Notably, a recent study demonstrated a unique correlation between molecular miR-181b/d indicators and health-related quality of life scores, reaffirming previous findings [100]. In addition, research conducted by Fengming Lan and colleagues revealed that exosomal miR-301a exhibits potential in reflecting both cancer status and pathological variations in human glioma, positioning it as a promising candidate for diagnostic and prognostic glioma biomarkers [101].

Nonetheless, the researchers’ pursuit of in-depth investigations into the evaluation of extracellular vesicles for the diagnosis and prognosis of glioblastoma continues. Wei et al. reported elevated levels of both miR-28-5p and miR-1224-5p, which were linked to improved prognoses [102]. Another study involved the analysis of plasma exosomes from 124 glioma patients and 36 non-tumor controls, revealing significantly reduced exosomal miR-2276-5p expression in glioma patients [103]. Receiver operating characteristic (ROC) curve analyses were employed to assess the diagnostic sensitivity and specificity of miR-2276-5p in glioma, resulting in an impressive area under the curve (AUC) of 0.8107.

Additionally, by scrutinizing serum exosomal miRNA profile datasets from GBM patients and normal controls sourced from the Gene Expression Omnibus database, researchers uncovered that hsa-miR-1835p and hsa-miR-985p, along with hsa-miR-323p or hsa-miR-19b-3p, constituted a diagnostic signature capable of distinguishing GBM from controls, with the area under the ROC curve approaching 1 [104]. The expression levels of these miRNAs in serum exosomes hold promise as reliable indicators of tumor progression.

Exosomal miRNAs, owing to their stability, ease of collection and detection, as well as their selectivity for tissues and cells, emerge as attractive candidates for biomarkers. They can be applied effectively for precise classification and early diagnosis of malignant gliomas. Furthermore, exosomal miRNAs stand as suitable biomarkers for patients who may not be eligible for surgery or are experiencing a recurrence, given their characteristics. For further insights into the potential applications of miRNA and other biomarker candidates in recent years, please refer to Table A1.

### 4.2. CircRNA

MiRNAs have emerged as prominent players in diagnostic research in the current state of the field. However, ongoing studies have delved into the roles of other non-coding RNAs in the detection and prognosis of glioblastoma. CircRNAs, a recently discovered class of endogenous noncoding RNAs, exhibit high stability, abundance, and conservation. They have been demonstrated to significantly contribute to the pathophysiological processes and tumor microenvironment remodeling in diverse cancers. A study unveiled the crucial involvement of circNEIL3 packaged in exosomes in promoting gliomagenesis, malignant progression, and polarization of macrophage tumor-promoting phenotypes [86]. This underscores circNEIL3’s potential as a prognostic biomarker and therapeutic target for glioma.

Another study revealed the abundance of circARID1A in exosomes produced by GBM cells. CircARID1A’s role in regulating the migration and invasion of GBM through the miR-370-3p/TGFBR2 pathway suggests its potential as a serum biomarker for the disease [70]. In a comprehensive examination, circRNAs that were dysregulated in exosomes isolated from plasma samples of GBM were evaluated using circRNA microarray analysis. Multiple steps, including a training group and a validation group, were employed to determine if the abnormally expressed plasma-derived circRNAs could predict GBM in comparison to healthy individuals. The exosome-derived hsa_circ_0055202, hsa_circ_0074920, and hsa_circ_0043722 were identified as having excellent potential as biomarkers for GBM [105].

Furthermore, circASPM, circFOXM1, and circNDC80 have been shown to contribute to GBM development, suggesting their potential as novel therapeutic targets and predictive biomarkers [106,107,108].

### 4.3. LncRNA

Long non-coding RNAs (lncRNAs) have diverse roles depending on their cellular location. They can influence transcriptional control, mRNA splicing in the nucleus, and mRNA stability and protein function in the cytoplasm [109]. In the past decade, it has become increasingly evident that long non-coding RNAs (lncRNAs) exert significant influence over a wide array of pathophysiological processes. These include the regulation of cell cycles, cell differentiation, and the innate immune response [110,111]. Moreover, disruptions in lncRNA expression have been observed in various human malignancies, gliomas among them, and have been associated with cancer staging and grading. LncRNAs can act as either tumor suppressors or promoters of tumorigenesis, underscoring their pivotal role in cancer biology [112]. High levels of lncSBF2-AS1 in serum exosomes have been linked to poor response to TMZ treatment in GBM patients, suggesting its potential as a diagnostic marker for therapy-resistant GBM [113]. Another study found that elevated levels of HOTAIR in serum-derived exosomes distinguished GBM patients from controls with high sensitivity and specificity [114]. Exosomal lncRNAs (HOTAIR, SOX21-AS1, and STEAP3-AS1) showed excellent concordance with tumor tissues, indicating their potential as predictors [115].

### 4.4. Protein

Exosome protein content associated with glioma malignancy plays a role in diagnosis and prognosis [116]. EGFRvIII, an oncogenic receptor variant, has been extensively studied and shows promise as a diagnostic marker for GBM [117]. After conducting further research, it has been determined that the semiquantitative exosome EGFRvIII polymerase chain reaction (PCR) detection assay in serum exhibits an overall clinical sensitivity of 81.58% and a specificity of 79.31% [118]. Recently, CD9-and CD81-positive EVs have been suggested as markers for tracking the radiation response of GBM tumors [119]. Other proteins like Notch1 and GOLPH3 also hold potential as diagnostic methods [120,121]. Currently, the prevailing method for diagnosing, treating, and predicting the outcomes of gliomas relies primarily on detecting alterations in the protein composition of tissues. Considering the potential of exosomal proteins as diagnostic and prognostic markers, employing exosomal proteins as diagnostic and prognostic indicators is a promising avenue of research.

In addition, continuous research on DNA and mRNA from exosomes suggests their potential as biomarkers for assessing glioma status [122,123]. Recent studies have introduced a minimally invasive procedure referred to as “liquid biopsy”, which is not only rapid and cost-effective but can also be applied at earlier stages of tumor development, even before macroscopic visibility, facilitating continuous monitoring of tumor progression [124]. This method involves identifying glioblastoma-specific exosomes in either blood or cerebrospinal fluid (CSF), offering a more comprehensive tumor characterization. A notable abundance of tumor exosomes is found in CSF, avoiding the blood–brain barrier, with limited interaction with non-tumor extracellular vesicles. Despite this, blood-derived exosomes are more accessible, leading to an ongoing debate about the preferred sample for GBM diagnosis [125].

Analyzing exosome samples and making predictions regarding therapeutic responses and prognosis provides insights into the type, origin, differentiation, and genotype of malignant gliomas. The cargo of exosomes, including DNA, mRNAs, proteins, and noncoding RNAs, has contributed to the identification of potentially significant biomarkers applicable to clinical diagnosis. Given the challenges of obtaining sufficient tissue samples for diagnosis, integrating clinical characteristics with genetic information from exosomes can lead to more precise diagnostic outcomes.

## 5. Exosomes in Glioma Treatment

In recent years, research on exosomes for GBM treatment has yielded promising results as scientists gain a deeper understanding of their components’ roles. This research can be broadly categorized into two main directions:

### 5.1. Utilization of Inherent Characteristics of Exosomes

Exosomes exhibit characteristics influenced by the organ and tissue of their origin, granting them distinct properties, including tropism to specific organs and uptake by particular cell types [24]. Notably, exosomes derived from human-umbilical-cord-derived mesenchymal stem cells (hUC-MSCs) have shown promise in partially inhibiting tumor growth by modulating the miR-10a-5p/PTEN pathway. Research conducted by Hao et al. suggests that hUC-MSCs-derived exosomes may offer alternative strategies for treating glioma [126]. Furthermore, exosomes derived from rat bone marrow mesenchymal stem cells (rBMMSCs) have demonstrated their potential as a standalone treatment for glioblastoma (GBM), independent of additional therapies or as a drug delivery system, as revealed in recent investigations [127]. Another study highlighted that exosomes derived from LPS/INFγ-triggered microglial cells have the ability to induce phenotypic shifts in tumor-associated myeloid cells. This shift is characterized by the upregulation of genes related to inflammation and leads to the inhibition of glioma growth [128]. Additionally, the incorporation of focused ultrasound has been shown to enhance the efficacy of exosome accumulation in gliomas, thereby improving the effectiveness of the treatment course [129].

Exosomes produced by cancer cells play a crucial role in driving overall tumor progression [82]. Consequently, there is ongoing research into methods for eliminating or suppressing their production, often considered as supplementary therapies. One noteworthy target in this effort is Munc 13-4, a Ca^2+^-dependent soluble N-ethylmaleimide-sensitive factor attachment protein (SNAP) receptor and Rab-binding protein. Studies have demonstrated that Munc 13-4 serves as the key regulator for the Ca^2+^ ion-mediated exosome release pathway in various cancer cell lines (although not specifically in glioma cells). Thus, targeted depletion of Munc 13-4 offers the potential to suppress oncogenic exosome secretion and hinder tumor progression [130]. Furthermore, the secretion of exosome-related PTRF can be controlled through the ubiquitination of PTRF by UBE2O, leading to a downregulation of exosome release [131]. Boosting UBE2O expression in cells holds promise as a novel approach for glioma treatment. In a similar vein, proton pump inhibitors (PPIs) and neutral sphingomyelinase 2 (nSMase2) are potential targets for reducing exosome secretion [132,133].

Exosome-mediated immunotherapy holds promise as an effective treatment approach, leveraging the unique properties of exosome donor cells [134]. Dendritic cells (DCs) stand out as highly efficient antigen-presenting cells, capable of eliciting potent immune responses. In preclinical studies, DC-derived exosomes have demonstrated superior anticancer efficacy compared to traditional DC vaccinations. This superiority arises from their enhanced immunogenicity and increased resistance against immunosuppressive factors [135,136]. Ning et al. made a significant discovery by demonstrating that DC-derived exosomes, loaded with chaperone-rich cell lysates and modulating Cbl-b and c-Cbl signaling, can trigger more robust antitumor T-cell immune responses [137]. Additionally, Liu et al. found that tumor-derived exosomes containing α-Galcer are effective for DC-based vaccinations. They employed α-Galcer-activated invariant NK-T cells as a cellular adjuvant, breaking immunological tolerance and generating antigen-specific cytotoxic T-cell (CTL) responses against GBM cells [138]. Furthermore, a MUC1 glycopeptide antigen was recently linked to dendritic-cell-derived exosomes to create a promising anticancer vaccine candidate. This construct not only increased cytokine production in vivo but also induced high MUC1-specific IgG antibody titers with strong binding affinity for MUC1-positive tumor cells [139]. Exosome-mediated immunotherapy has gained significant attention recently and exhibits remarkable potential as a treatment option for gliomas [140].

### 5.2. Exosomes as Delivery Systems

Exosomes, owing to their biocompatibility, stability, and diminutive nano-scale dimensions, have been engineered to serve as effective carriers for therapeutic compounds, demonstrating promising initial success. 

#### 5.2.1. Chemotherapy Drugs as Exosome Payloads

Exosomes can serve as efficient carriers for delivering chemotherapy medications into the brain and glioma site, overcoming the restrictions of the blood–brain barrier. Studies have utilized brain-endothelial-cell-isolated exosomes loaded with paclitaxel and doxorubicin, demonstrating their potential to inhibit GBM growth [141]. Similarly, coating doxorubicin-loaded nanoparticles with brain-endothelial-cell-derived exosomes allows for effective drug delivery to GBM cells, extending the survival of GBM-bearing animals [142]. Neutrophil exosomes, due to their inflammation-driven nature and BBB-crossing ability, have also been employed for transporting doxorubicin, effectively targeting infiltrating GBM cells [143].

Combining modified exosomes with other drugs, such as temozolomide, further enhances the efficacy of GBM therapy. Modified exosomes loaded with paclitaxel, created from embryonic stem cells and conjugated with a tumor-targeting peptide, have shown improved targeting and therapeutic effectiveness against GBM cells [55]. In addition, exosomes were loaded with superparamagnetic iron oxide nanoparticles (SPIONs) and curcumin (Cur), and click chemistry was employed to attach a neuropilin-1-targeted peptide to the exosome membrane, thus creating glioma-targeting exosomes with combined imaging and therapeutic capabilities [144]. Drawing inspiration from brain-targeting exosomes, Wang and colleagues reported the exploration of biomimetic nanovesicles [145]. These were engineered through membrane fusion between blood exosomes and tLyp-1 peptide-modified liposomes, demonstrating potential for brain-targeted drug delivery. These examples underscore the potential of exosomes in drug delivery against brain tumors. More research and details are shown in Table A2.

#### 5.2.2. Nucleotide Drugs as Exosome Payloads

Nucleotide-based gene therapy holds great promise in clinical applications. Exosomes can potentially deliver mRNA, noncoding RNA, and antisense oligonucleotides (AON) to inhibit glioma. Studies have shown that exosomes loaded with miR-124a or miR29a-3p can significantly reduce the vitality and clonogenicity of GSCs [146,147]. Targeted modification of exosomes further enhances their delivery efficiency [51]. These results demonstrate the potential of exosomes as delivery systems for anti-tumor nucleotide drugs, indicating their potential for therapeutic use. Before these treatment approaches can be implemented in clinical settings, further extensive research is needed.

## 6. Future Directions and Conclusions

While the initial findings are promising, there are critical steps that need to be taken before exosomes can be integrated into clinical practice. A significant challenge lies in the lack of standardized protocols for exosome isolation, analysis, and methods for loading and modification. Understanding the precise role and mechanism of exosomes in intercellular communication is foundational to exploring their potential as diagnostic and therapeutic tools. This necessitates further research to elucidate detailed mechanisms and validate additional biomolecules as diagnostic and therapeutic strategies. Machine learning algorithms, based on artificial intelligence, could prove invaluable in identifying potential biomarkers from secretions in various malignant tumors and in predicting diagnoses and prognoses of pathological conditions.

Moreover, the integration of hybrid inorganic/organic nanoparticles with exosomes shows promise in enhancing drug-delivery efficiency and controlled release, addressing challenges such as inefficient separation procedures, characterization difficulties, and the lack of specific biomarkers associated with exosomes. Finally, more extensive clinical trials are essential to validate the efficacy of exosomes as potential standard treatment agents and biomarker tools for gliomas and other cancers, despite a limited number of trials having been carried out in certain disease areas [148]. In the future, with the realization of the clinical application of exosomes in glioblastoma, continued exploration may focus on personalized exosome-based treatments tailored to individual patient profiles.

Exosomes represent a novel mode of cell communication that plays a pivotal role in influencing various aspects of gliomas, including proliferation, invasion, angiogenesis, immune escape, and therapy resistance. They hold great promise as candidates for glioma diagnosis and prognosis due to their rich content of tumor-specific bioactive molecules. Additionally, their ability to naturally cross the blood–brain barrier, biocompatibility, low immunogenicity, and extensive surface-engineering capabilities make them a focus for research as carriers for chemotherapeutic and nucleotide drugs aimed at suppressing gliomas.

Founded on the aforementioned conceptual framework, this review outlines the capability of exosomes to breach the blood–brain barrier and their role in glioma progression. It adds to a summary of recent advancements in the diagnosis, prognosis, and treatment of glioma, underscoring a critical direction for exosome development. The aim is to provide researchers in relevant fields with a comprehensive and clear framework for applying exosomes in glioma, enabling them to grasp the latest research trends and future development directions.

## Figures and Tables

**Figure 1 cancers-16-00823-f001:**
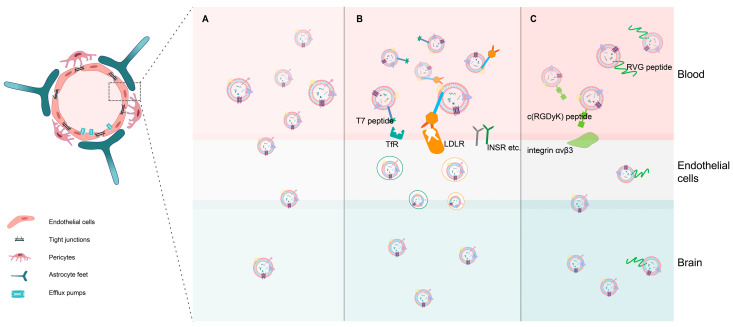
Strategies of exosomes for crossing the BBB. (**A**) Exosomes possess inherent BBB permeability. (**B**) Exosomes can hijack receptor-mediated transcytosis (RMT) by specific ligand-receptors binding to control transcytosis. (**C**) Exosomes can be modified with membrane-protein-binding peptides or neurotropic-virus-derived peptides to achieve effective delivery.

**Figure 2 cancers-16-00823-f002:**
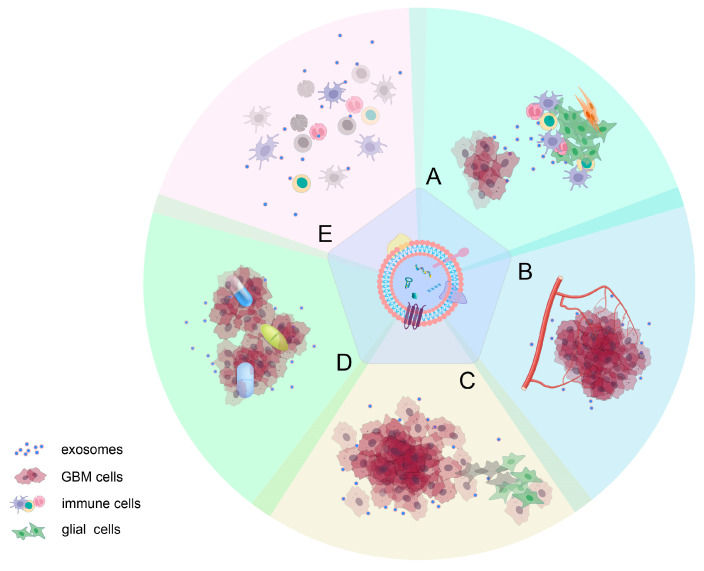
Roles of GDEs in glioma progression: (**A**) inducing changes in the tumor microenvironment; (**B**) mediating angiogenesis; (**C**) influencing proliferation and invasiveness of gliomas; (**D**) contributing to drug resistance; and (**E**) suppressing immune responses. It is important to note that the effects of active molecules on tumor progression are often not isolated but rather have a wide-ranging impact on various aspects of glioma development.

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
