# Peer review of "Exosomes in Glioma: Unraveling Their Roles in Progression, Diagnosis, and Therapy"

_cancers, 2024, doi:10.3390/cancers16040823_

Round 1

Reviewer 1 Report

Comments and Suggestions for Authors

The ms by Yang et al. recapitulates the current knowledge on exosomes in gliomas, a current and developing field of research. Such a summary is most welcomed.

However, I would have some points for review.

The ms deals with exosomes in gliomas, a collection of entities. Therefore any focusing on one type (like some detailed descriptions of GBM - lines 46-58) is out of subject unless it is extended to other glioma types.

Moreover, the further part of the ms contains a part focused on GBM - the specific GBM information can be located there.

Paragraph 2.2 concerning isolation methods - in the first part enumerating those methods I cannot see relevant references. It is also not specified from what kind of material exosomes are isolated.

Paragraph 2.3 on crossing BBB - a very important part of the ms structure - should be definitely more precise, any table/graph/enumeration of the mentioned affinities or crossing mechanisms should be included, not only single examples.

I could not define the added value of Figure 2. Presents no specific process, visually adds nothing, and can be replaced by text description or extended table.

Figure 3 is very nice, but too general and unrelated to the subject of the ms.

Overall, there's a lot of potential in the text for summarizing into tables/schemes/figures. The authors should present those not-single examples in a more visual-friendly form for better subject presentation.

Comments on the Quality of English Language

There are some spelling/stylistic/grammatical/punctuation mistakes - needs one thorough reading.

Especially the colloquial style should be corrected to be more neutral and facts presenting.

Reviewer 2 Report

Comments and Suggestions for Authors

In the submitted manuscript authors review aspects connected with the role of exosomes in glioma. While the topic is interesting and generally nicely written, I have some major comments about this work.

First, this review lacks the methodology sections. While this review is, at least partially, a systematic review revie – due to the presence of tables A1 and A2, there are no information on how actually the choice of the article has been done. At least the study design and search strategy should be presented, together with the study selection and criteria. Besides, Cancers highly recommend preparing the review in accordance with the PRISMA rules, which has not been done. Therefore, the Authors should include the flowchart based on the PRISMA statement.

Line 80, at this point I miss a short paragraph describing the aim of this review and its construction.

Line 99, here, it should be sated if the EVs are found only in human or in the other organisms as well

Line 116, “many chemotherapeutics”, please list some most important examples

Figures 1-3, since this is a review, I have to ask, were those figures created by the Authors or were they reproduced from the earlier works (which would be fine, too)? Also, the font is very small, it is extremely hard to read some words.

Line 164, can the authors comment on this statement? Does it mean that in the future, the non-modified exosomes from the healthy cells can be possible used as a drug? Here, a reference to the 5.1. sections should be also made.

Line 240, what about the pharmacotherapy?

Lines 399, 401, it should be “Ca2+

Reviewer 3 Report

Comments and Suggestions for Authors

This study explores the promising role of exosomes in glioblastoma (GBM) treatment through two main directions: leveraging the inherent therapeutic characteristics of exosomes, particularly those derived from specific cell types, and utilizing exosomes as effective delivery systems for chemotherapy drugs and nucleotide-based gene therapies. The review emphasizes the potential of exosomes for diagnosis, prognosis, and personalized treatment in GBM, outlines critical steps for clinical integration, and highlights the need for standardized protocols and extensive clinical trials for validation.

Section 1:

1. While the transition to nanocarrier systems is well-executed, a more explicit link between the challenges of current treatments and the need for innovative approaches, such as nanotechnology, could strengthen the narrative.

2. The introduction could benefit from a brief mention of potential future perspectives or the significance of exploring novel therapeutic approaches. This would provide a sense of direction for the reader.

3. While the focus is on nanocarrier systems and exosomes, acknowledging briefly any alternative emerging therapies in the field of glioma treatment could add completeness to the introduction.

Section 2

1. Consider integrating brief descriptions or references to Figure 1 within the text to guide readers seamlessly through the strategies of exosomes for crossing the BBB.

Section 3

Ensure that citations are provided for recent studies mentioned in the section, especially when introducing specific miRNAs, circRNAs, or proteins associated with exosomes. This ensures transparency and allows readers to access the original research.

Section 4

1.Consider breaking down lengthy paragraphs into smaller, more digestible sections for improved readability. This can enhance the overall flow of the content.

Reviewer 4 Report

Comments and Suggestions for Authors

This topic is very interesting, look at these points to improve the paper:

- Lines 77-79. Introduction section. It is not clear what is the purpose of this paper. Revise this part.

- Lines 435-438. Wang and colleagues reported "a biomimetic nanoplatform for increased BBB transcytosis into brain parenchyma". Discuss more here at this point.

- Lines 128-30: "For example, neural stem cell-derived EVs demonstrated enhanced CNS distribution compared to mesenchymal stem cell-derived EVs in a mouse stroke model " What do the authors mean with this example? Improve this part or remove it.

- Lines 248-250: " While some potential biomarkers have been found in the serum and cerebrospinal fluid (CSF) of glioma patients, their clinical utility has not progressed significantly" This sentence seems to conflict with the first part of the paper. Revise it.

- Lines 214-216: "Moreover, research has highlighted the potential of the exosome-mediated circWDR62 and macrophage migration inhibitory factor (MIF) in increasing TMZ resistance in glioma, suggesting their value as prognostic biomarkers" Highlight the role of the tumor macrophages also in recurrent GBM. Look at these one:  --  doi: 10.3390/neurolint15020037 --  PMCID: PMC10204554 -- doi: 10.1186/s12935-024-03225-4

- the conclusion must be able to bring new considerations to the reader. Improve the conclusion: what does this article add new to previous literature? Although it is a review, the conclusion can be improved.

- Figure 3 is difficult to read and it not clear why authors put figure 3D. Revise it

Comments on the Quality of English Language

Minor editing of English language required.

Round 2

Reviewer 2 Report

Comments and Suggestions for Authors

The Authors have improved and corrected their manuscript. This version can be accepted.

Reviewer 4 Report

Comments and Suggestions for Authors

The authors solved all my criticisms.